# Structure–function insights reveal the human ribosome as a cancer target for antibiotics

Alexander G. Myasnikov[1,2,3,4,*], S. Kundhavai Natchiar[1,2,3,4,*], Marielle Nebout[5,6,*], Isabelle Hazemann[1,2,3,4], Véronique Imbert[5,6], Heena Khatter[1,2,3,4,†], Jean-François Peyron[5,6] & Bruno P. Klaholz[1,2,3,4]

Many antibiotics in clinical use target the bacterial ribosome by interfering with the protein synthesis machinery. However, targeting the human ribosome in the case of protein synthesis deregulations such as in highly proliferating cancer cells has not been investigated at the molecular level up to now. Here we report the structure of the human 80S ribosome with a eukaryote-specific antibiotic and show its anti-proliferative effect on several cancer cell lines. The structure provides insights into the detailed interactions in a ligand-binding pocket of the human ribosome that are required for structure-assisted drug design. Furthermore, anti-proliferative dose response in leukaemic cells and interference with synthesis of c-myc and mcl-1 short-lived protein markers reveals specificity of a series of eukaryote-specific antibiotics towards cytosolic rather than mitochondrial ribosomes, uncovering the human ribosome as a promising cancer target.

[1] Centre for Integrative Biology (CBI), Department of Integrated Structural Biology, IGBMC (Institute of Genetics and of Molecular and Cellular Biology), 1 rue Laurent Fries, Illkirch 67404, France. [2] Centre National de la Recherche Scientifique (CNRS) UMR 7104, Illkirch, France. [3] Institut National de la Santé et de la Recherche Médicale (INSERM) U964, Illkirch, France. [4] Université de Strasbourg, Strasbourg, France. [5] INSERM, U1065, Centre Méditerranéen de Médecine Moléculaire (C3M), Nice, France. [6] Université de Nice-Sophia Antipolis, UFR Médecine, Faculté de Médecine, Nice, France. * These authors contributed equally to this work. † Present address: European Molecular Biology Laboratory (EMBL), Structural and Computational Biology Unit, Meyerhofstrasse 1, 69117 Heidelberg, Germany. Correspondence and requests for materials should be addressed to J.-F.P. (email: peyron@unice.fr) or to B.P.K. (email: klaholz@igbmc.fr).

The ribosome is the molecular machinery at the heart of protein synthesis, a highly regulated activity which is tightly connected with cell activation and proliferation, with many steps controlled by both proto-oncogenes and tumour suppressors. Elevated protein synthesis rates and up-regulated ribosome biogenesis are characteristic hallmarks of cancer cells because these highly proliferating cells have a vital need for new cellular constituents[1]. The importance of exacerbated protein synthesis and ribosome function in cancer is illustrated by the participation of the Myc oncogene in stimulating expression of initiation/elongation factors and ribosomal proteins during cell transformation[2]. About half of the currently existing antibiotics target the bacterial ribosome by interfering with initiation, elongation, termination and other regulatory mechanisms[3,4]. While some antibiotics are known for their anti-tumoral activities, the mechanism of action and target definition often remain poorly understood, including whether mitochondrial or cytosolic ribosomes are the target. For example, homoharringtonine (Omacetaxine) was screened as an alkaloid with anti-tumoral properties and was shown later to affect protein synthesis, it now has become the first approved drug against chronic myelogenous leukaemia[5,6]. Nevertheless, targeting the human ribosome has not been envisaged with respect to drug design yet, and dedicated work is required to address the problem of targeting an essential cellular function in the human body and potential side effects if entirely blocked. Indeed, it should in principle be possible to differentially modulate protein synthesis activity of the human ribosome at sufficiently low ligand doses and thereby primarily target strongly proliferating cells such as cancer cells. Moreover, because of their high protein synthesis rate, cancer cells develop addictions and are expected to be highly sensitive to their inhibition, compared with normal untransformed cells. T-cell Acute Lymphoblastic Leukaemia (T-ALL) and T-cell Lymphoblastic Lymphoma (T-LL), which are highly aggressive cancers with frequent relapses after initial treatment and are refractory to currently available drugs[7], display a pathological addiction to essential amino acids and protein synthesis[8]. Until recently, it was not possible to envision studying the molecular and structural basis of ligand actions on the human ribosome. This has now changed with our recently obtained first high-resolution structure of the human ribosome using advanced cryo-electron microscopy (cryo-EM)[9]. We decided to analyse a eukaryote-specific inhibitor of protein biosynthesis, cycloheximide (CHX), which is produced by the bacterium *Streptomyces griseus* and is widely used *in vitro* for biomedical research on protein synthesis in eukaryotic cells. A crystal structure of CHX bound to the yeast ribosome has revealed the location of the binding site on the ribosome suggesting that CHX and the 3′ CCA end of the exit (E) site transfer RNA (tRNA) share a common binding region at the E-site[10], but the detailed mechanism of action remained to be addressed. Moreover, it is important to conduct structural analyses on the human ribosome rather than on any related model system (bacteria or yeast) to allow a precise analysis of drug interactions with the correct medical target for applications in human health.

We have now determined the first human 80S ribosome structure with a ligand. The structural comparison of this ligand complex with our previously published apo 80S complex[9] reveals the molecular mechanism which is based on a dynamic ligand-induced active release of the E-site tRNA. Furthermore and importantly, we provide evidence for the anti-proliferative activity of CHX which extends to a series of ligands exhibiting a marked specificity towards the cytosolic ribosome, thus establishing the human ribosome as a promising cancer target. This structure and function analysis performed on the human ribosome using a variety of drug candidates provides important insights for the development of new antibiotics.

## Results

**Structural analysis of the human 80S/CHX complex.** Human ribosomes were prepared as described[11], incubated with CHX to form the complex and the structure was determined by single particle cryo-EM and refined to an average resolution of 3.6 Å (that is, local resolution features go beyond this value; see Methods and Supplementary Figs 1 and 2). Atomic model building and model refinement against the cryo-EM map was done using the previous structure as a starting point[9]. Particle sorting and three-dimensional (3D) structure classification revealed several species in the sample, notably two main 80S ribosome structures in absence or presence of endogenous E-site tRNA (see Methods; Supplementary Fig. 3; Fig. 1). The tRNA-containing 80S complex exhibits no density for CHX, and the tRNA position and conformation are virtually identical to those seen previously in the human 80S structure in the absence of CHX[9]. In contrast, the structure lacking the E-site tRNA shows an additional well-defined density at the bottom of the E-site corresponding to CHX (Fig. 2). The cyclohexyl moiety of the ligand is found in vicinity to bases G4370 and G4371 of the 28S ribosomal RNA (rRNA) of the 60S subunit, while the piperidine-2,6-dione (glutarimide) moiety protrudes into a cavity next to the G91 region of the 28S rRNA (Fig. 2c). The overall binding site corresponds to that observed in the yeast ribosome[10] but the ligand is slightly rotated, has the piperidine-2,6-dione moiety flipped and exhibits an additional interaction with the 2-keto group of C4342 possibly related with the presence of a $Mg^{2+}$ ion which is absent in the yeast ribosome complex (Fig. 2d). Specific interactions in the ligand pocket (Fig. 2c) are achieved through (i) the piperidine-2,6-dione group which is in hydrogen-bonding distance to U4340, G91 (2′-OH), C92 (phosphate) and the associated $Mg^{2+}$ ion, (ii) the carbonyl group of the cyclohexyl moiety which can hydrogen bond with C4341 and (iii) the hydroxyl group in the linker that points towards C4342. These interactions are strikingly different from those of the 3′-terminal adenine A76 of the tRNA CCA end (Fig. 2e), in particular for the cyclohexyl moiety. While A76 intercalates between the 28S rRNA bases G4370/G4371 and interacts with C4341 in a non-standard manner to form a helical RNA segment (Fig. 2e and top view in Fig. 2f), CHX does not intercalate (Fig. 2c,g); instead, it requires additional interactions through the piperidine-2,6-dione moiety (Fig. 2c) to compensate for the numerous interactions of the tRNA in the acceptor stem region (Fig. 1d). This explains how a small ligand like CHX (281 Da) can compete for the binding of a large ribosomal ligand (tRNA, 25 kDa) even though it overlaps only marginally with the tRNA considering its large size (Figs 1c,d and 2g). These observations illustrate the dynamic nature of ligand binding and tRNA release (which occurs even at 4 °C during the ligand incubation, see Methods), involving conformational changes at the ribosome level, notably helices H77/78 of the L1 stalk region which becomes disordered (Fig. 1): CHX induces release of tRNA, and inversely once CHX is bound it prevents the tRNA from re-binding; moreover, during the elongation phase, it would block tRNA translocation from the P- to the E-site. This first structure of a ligand complex with the human ribosome provides unprecedented insights into the detailed interactions in a typical ligand-binding pocket that are required for structure-assisted drug design. It suggests, in particular, that ligands could be synthesized with an aromatic extension to imitate the intercalation of A76 (Fig. 2h,g). Furthermore, while interactions with rRNA elements are seen on one side of the ligand only, the

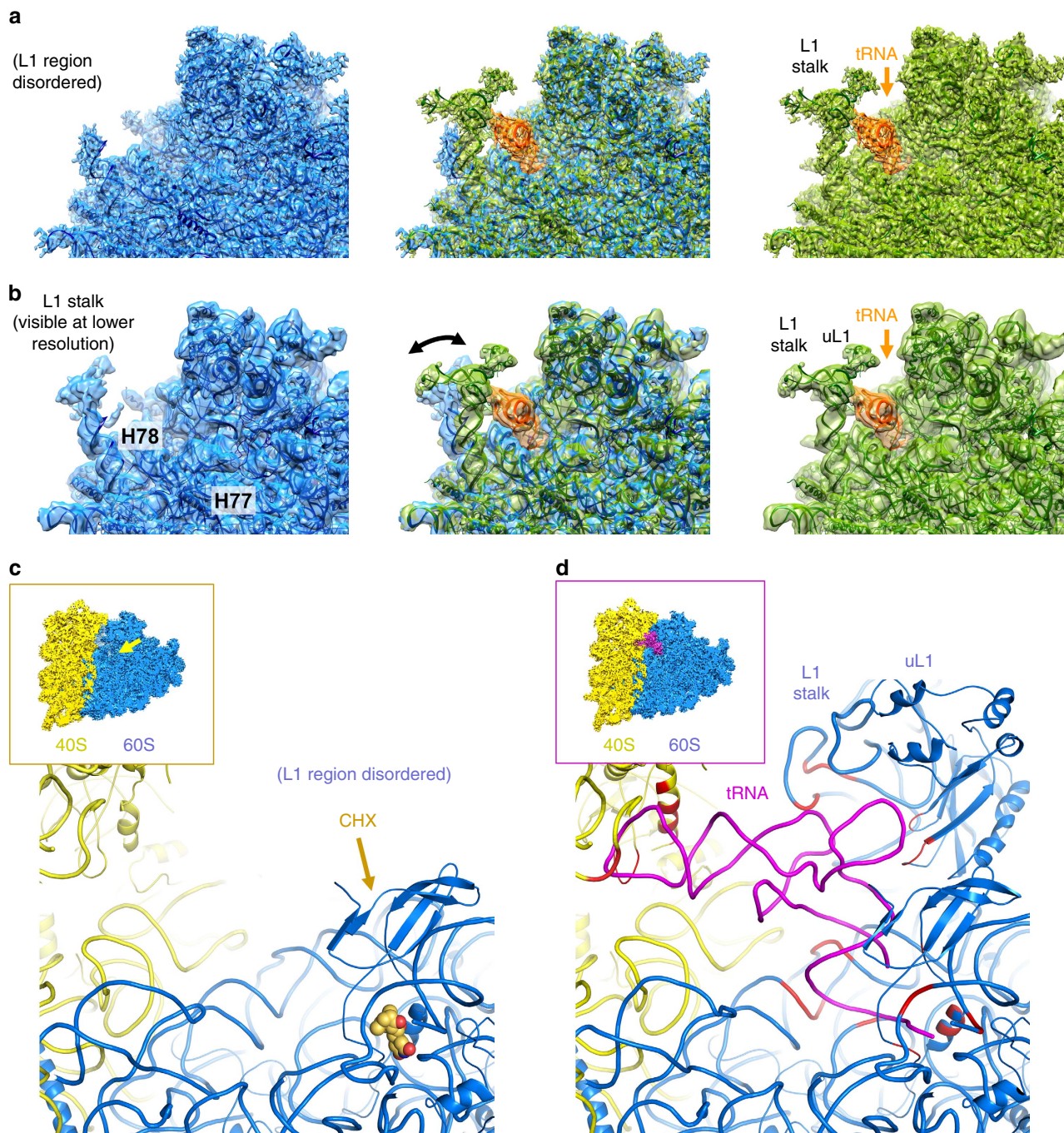

**Figure 1 | Structure of the human 80S ribosome with the eukaryote-specific CHX ligand.** (**a**,**b**) Comparison of the 80S ribosome complexes with the CHX ligand or with E-site tRNA bound in the sample separated by particle sorting, showing the conformational changes of the L1 region on the ribosome occurring upon CHX binding and tRNA release (left, 80S/CHX complex; right, 80S with E-site tRNA, middle superposition; cryo-EM maps and corresponding atomic models are shown in blue and green respectively). The structural disorder of the L1 region upon CHX binding and tRNA release is illustrated by the fact that it is better visible in cryo-EM maps filtered to 5 Å (**b**) compared with the full-resolution maps (**a**). (**c**) Overall binding region of CHX on the human 80S ribosome (backbone representation; CHX is shown with van der Waals spheres); the L1 region is disordered because of the absence of E-site tRNA. The inset shows the overall cryo-EM map sectioned at the level of the ligand-binding pocket (arrow; 60S and 40S subunits are annotated). (**d**) Structural comparison with the human 80S ribosome containing E-site tRNA[10]. View on the exit (E) site tRNA including the rRNA components interacting with the tRNA (marked in red) which are well-structured including the L1 region.

other side facing the Lys53-Phe56 region of ribosomal protein eL42 could still be exploited in a modified compound to form new interactions (Fig. 2h).

**Functional analysis of antibiotics.** The above findings on the mechanism of action of CHX on the human ribosome prompted

us to test the hypothesis of whether affecting protein synthesis by eukaryote-specific (E-) antibiotics may result in anti-proliferative effects in leukaemic cells. We recently demonstrated that T-ALL cells display an addiction to essential amino acids[8] that supports the constitutive activation of mammalian Target of Rapamycin Complex 1, an important inducer of protein synthesis and cellular growth[12]. We first used a representative murine T-ALL model

generated by the T-cell-specific deletion of the Phosphatase and TENsin homologue (PTEN) tumour suppressor (tPTEN − / −) (ref. 13). Transformed tPTEN − / − cells (see Methods) display an intense ribosome biogenesis, likely a consequence of

c-myc amplification[2]. Using tPTEN − / − leukaemic cells, we observed that a panel of E-antibiotics (anisomycin, CHX, deoxynivalenol, homoharringtonine and verrucarin-A) exert a dramatic dose-dependent decrease in leukaemic cell viability

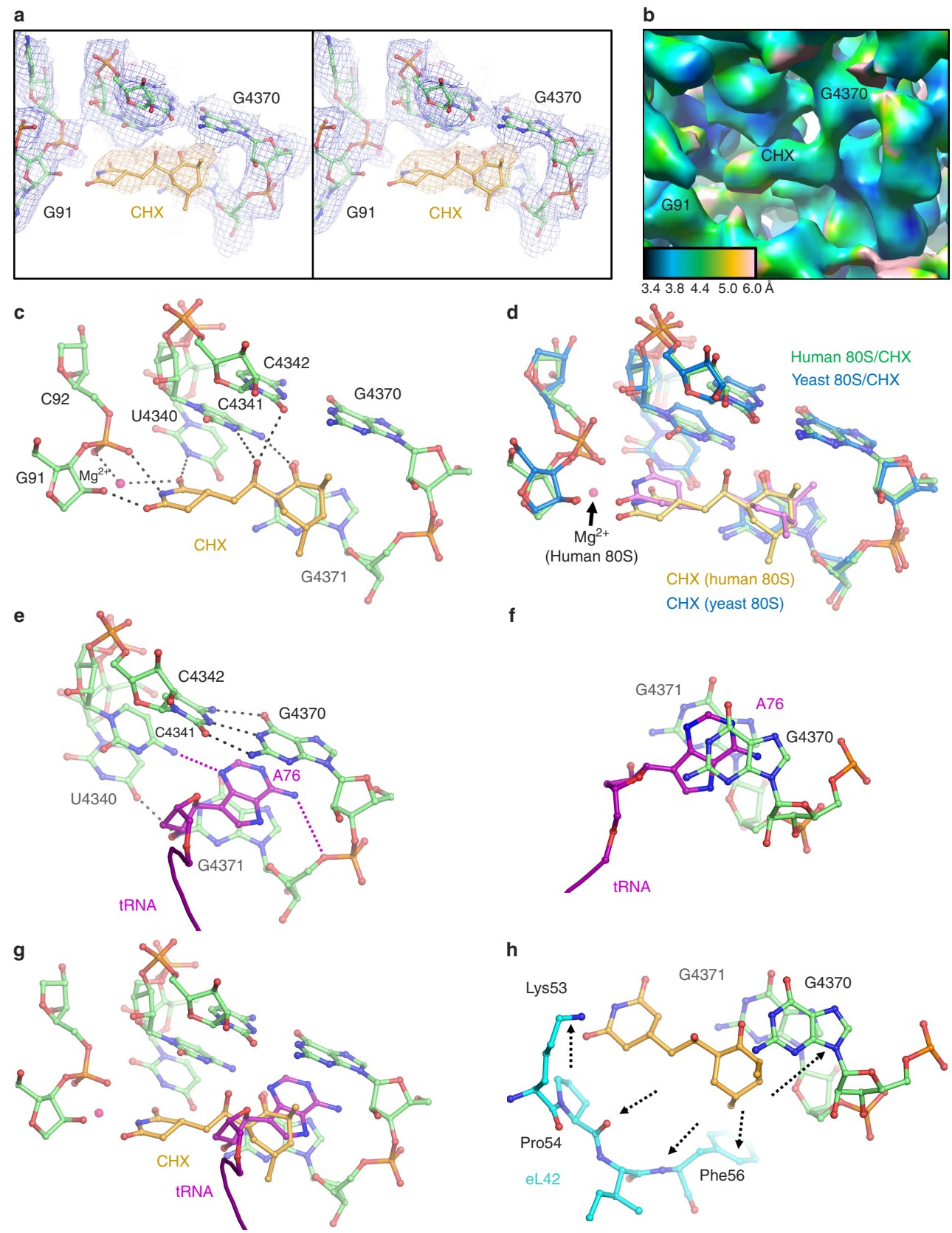

measured through a mitochondrial activity assay using the WST1 tetrazolium salt (Fig. 3a, top); this is consistent with the anti-chronic myelogenous leukaemia activity of homoharringtonine[5,6]. $IC_{50}$ values range from 15.0 nM for homoharringtonine to 479 nM for deoxynivalenol, with 248 nM for CHX (Supplementary Fig. 4). The loss in viability was associated with an important dose-dependent decrease in cell numbers (Fig. 3a, middle) and to an induction of apoptosis, evidenced by flow cytometry analysis (Fig. 3a, bottom; see $IC_{50}$ values in Supplementary Fig. 4). When tested on normal peripheral blood lymphocytes, anisomycin and verrucarin-A displayed some toxicity for normal cells (36% and 24% increase in cell death, respectively) but only at very high doses (30 and 1 μM, respectively), while CHX did not trigger cell death in normal cells compared with leukaemic cell lines (Supplementary Fig. 5). These results are explained by the existence of a protein synthesis addiction in leukaemic but not in normal cells.

We next addressed the specificity of cytosolic versus mitochondrial ribosomes, the latter being more related to, although different from, prokaryote ribosomes that are usually considered as the main medical target. Strikingly, we found that four prokaryote-specific (P) antibiotics (gentamicin G418, kanamycin, streptomycin and tetracycline) did not affect cell viability, proliferation nor induced death of tPTEN − / − cells (Fig. 3b). The anti-leukaemic potential of the E-antibiotics CHX, homoharringtonin and verrucarin-A was further evidenced by a dose-dependent decrease in viability (Fig. 3c, top) and parallel increase in cell death (Fig. 3c, bottom) on a panel of human leukaemic cell lines: T-ALL (Jurkat) and B-ALL (NALM-6, 697) and Acute Myeloblastic leukaemia (Kasumi-1, U937) (see $IC_{50}$ values in Supplementary Figs 6 and 7). By contrast, no anti-leukaemic effects of the P-antibiotics could be detected on these cell lines (not shown).

The specificity in the mode of action of these different classes of antibiotics was then investigated at the molecular level. We analysed translation of distinct cellular markers, the c-myc and mcl-1 proteins which are synthesized by cytosolic ribosomes, and cox-2, a component of the mitochondrial respiratory complex IV translated by mitoribosomes. These proteins display a short half-life (<15 min) making them particularly sensitive markers for protein synthesis rate variations. Strikingly, the levels of c-myc and mcl-1 were greatly reduced by E-, but not P-antibiotics (Fig. 3d). Conversely, cox-2 protein levels sharply dropped following treatment with P-antibiotics and less strongly by E-antibiotics. At these relatively low concentrations, none of the antibiotics modified total protein synthesis as shown by a lack of effect on Hsp90 protein levels. The reduction in c-myc, mcl-1 and cox-2 protein levels is independent of their respective mRNAs levels which remained unaffected (Supplementary Fig. 8). While anti-proliferative activities of CHX and other antibiotics had been suggested previously[14,15] their mechanism of action and target

definition remained unclear. The present data now provide a detailed quantification and clarify the molecular mechanism of action over a whole series of ligands to demonstrate that eukaryote-specific antibiotics carry a strong anti-cancer potential and indeed target the human ribosome.

## Discussion

Taken together, the structural analysis of the 80S–CHX complex shows the feasibility of elucidating the molecular basis of ligand interactions with the human ribosome. Noteworthy, the flipping of the piperidine-2,6-dione moiety, the additional $Mg^{2+}$ ion and the additional H-bond opportunity with C4342 would not be predicted from the yeast ribosome crystal structure with CHX, illustrating the importance of studying drug interactions with the right medical target. Along with the findings on the anti-proliferative action of CHX and a series of eukaryote-specific antibiotics, this study provides the experimentally validated proof-of-principle that the human cytosolic ribosome is a promising target to consider for innovative anti-cancer therapies. Whether human ribosomes differ in normal and cancer cells is an interesting aspect to analyse in the future. Rather than blocking, it should indeed become possible to modulate protein synthesis activity of the human ribosome and thereby target strongly proliferating cells. This work thus opens many new opportunities in translational medicine and enables future studies of ribosome complexes with E-antibiotics in combination with structure-guided design to help developing new anti-cancer drugs with higher specificity and a lower general toxicity than those currently available.

## Methods

**Complex formation and structure determination.** Human 80S ribosomes were prepared from HeLa cells as described earlier[11]. A volume of 2.5 μl of freshly prepared human 80S ribosomes containing CHX (4-[(2R)-2-[(1S,3S,5S)-3, 5-dimethyl-2-oxocyclohexyl]-2-hydroxyethyl]piperidine-2,6-dione; 10-fold molar excess of CHX in $H_2O$, incubated 10 min on ice), diluted from 2 mg ml$^{-1}$ to 0.5 mg ml$^{-1}$, was applied to 300 mesh holey carbon Quantifoil 2/2 grids (Quantifoil Micro Tools, Jena, Germany) and flash-frozen as described[9]. Data were collected on the in-house spherical aberration (Cs) corrected Titan Krios S-FEG instrument (FEI, Eindhoven, Netherlands) operating at 300 kV acceleration voltage and at a nominal underfocus of $\Delta z = -0.4$ to $-3.0$ μm at a magnification of ×79,000, corresponding to 0.88 Å per pixels on the specimen level. Images were recorded using a back-thinned direct electron detector (Falcon II) 4096 × 4096 camera with dose fractionation (seven individual frames were collected, starting from the second one). Total exposure time was 1 s with a dose of 60 ē Å$^{-2}$ (or 3.5 ē Å$^{-2}$ per frame). Images in the stack were aligned using the whole image motion correction method described in ref. 16. Particles (159,000) were picked automatically using the in-house software gEMpicker[17] and the contrast transfer function of every image was determined using CTFFIND4 (ref. 18) in the RELION workflow[19]. For the first steps of refinement, images were coarsened by 2 (C2 images) and 4 (C4 images) using EMAN2. First, we applied 3D classification to remove bad particles (119,386 particles left), followed by two-dimensional (2D) classification to remove images with ice or noise (95,917 good particles; for 2D and 3D classifications see also ref. 20). After 3D refinement, we performed one more 3D classification (see workflow in Supplementary Fig. 3) to split rotated (24,714) and

**Figure 2 | Analysis of CHX interactions in the ligand-binding pocket of the human 80S ribosome.** (**a**) Cryo-EM map of the ligand-binding pocket including the CHX ligand (stereo representation). (**b**) Local resolution estimation of the CHX ligand and of the surrounding residues. (**c**) Details of the molecular interactions of CHX in the ligand binding pocket, the residues in hydrogen-bonding distance are indicated. The cyclohexyl moiety of the CHX ligand (on the left) interacts with G91 region, while the piperidine-2,6-dione moiety is located next to G4370 and G4371 of the 28S rRNA of the 60S subunit. (**d**) Comparison of the yeast and human ribosome structure with CHX, showing that the ligand is slightly rotated, has the piperidine-2,6-dione moiety flipped. Note the presence of a $Mg^{2+}$ ion which is absent in the yeast ribosome complex. (**e**) Detailed view of the interactions of the 3′-terminal adenine A76 of the tRNA with the 28S rRNA; same viewing angle as in panel **c**. The adenine intercalates between G4370 and G4371 of the 28S rRNA and forms a non-standard interaction with C4341 through the atom N3 acceptor position. Base pair hydrogen bonds are indicated for the neighbour nucleotides. (**f**) Intercalation of the 3′-terminal A76 of the tRNA between bases G4370 and G4371 (view along the base stacking, viewing from the top as compared with panel **e**). (**g**) The superposition of the CHX ligand (yellow) and the E-site tRNA (magenta) shows that they overlap only partially, while the main part of the tRNA extends further into the E-site (viewing angle as in **e**). (**h**) Illustration of the potential of structure-assisted drug design. The arrows indicate potential interaction sites in the vicinity (within 4–5 Å distance), notably the Lys53-Phe56 region of ribosomal protein eL42 and the possible intercalation between bases G4370 and G4371 which would imitate that of A76 of the tRNA (**f**, same viewing angle).

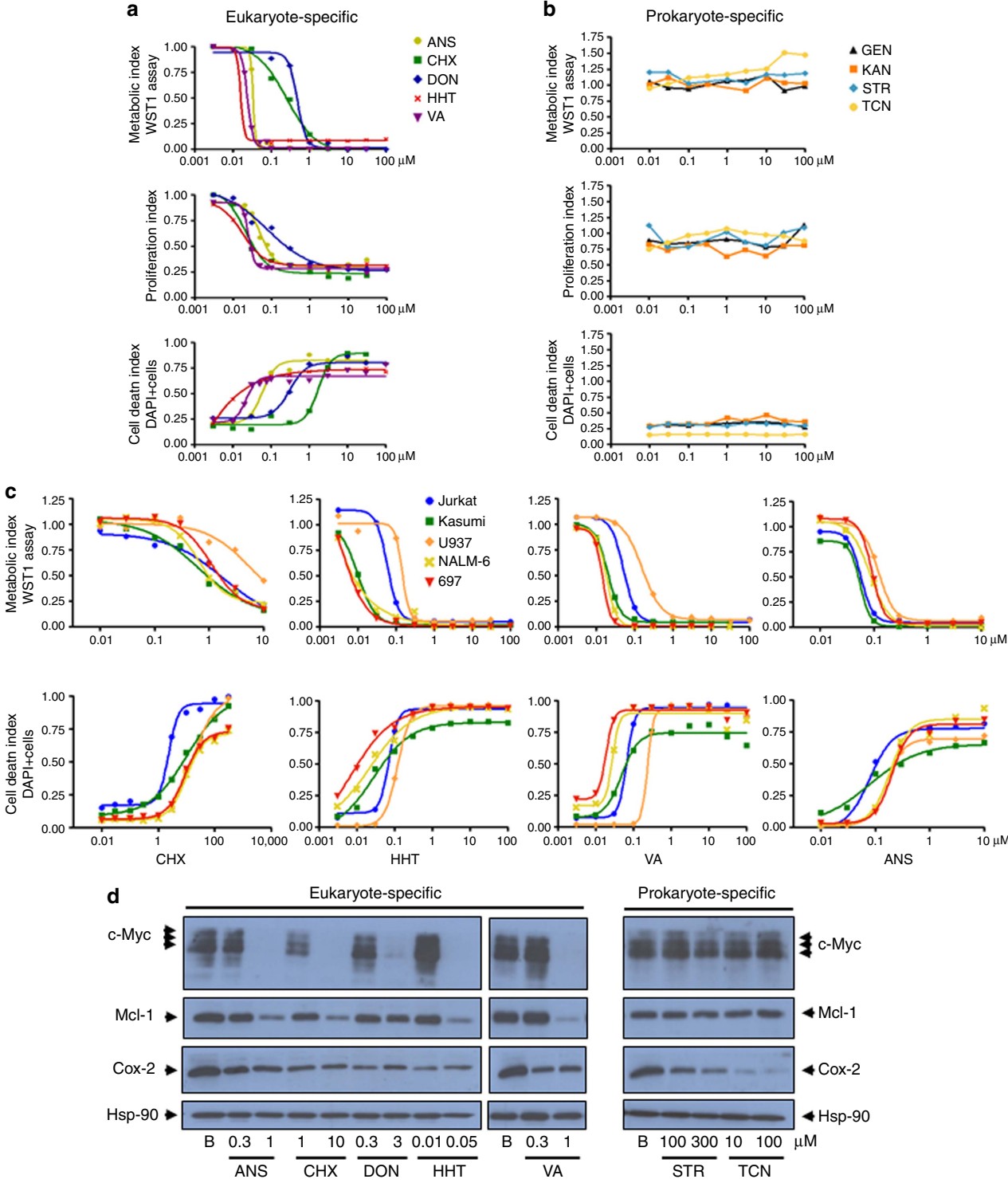

**Figure 3 | CHX and other eukaryote-specific but not prokaryote-specific antibiotics display anti-leukaemic activity.** (**a**,**b**) Eukaryote-specific antibiotics: anisomycin (ANS), cycloheximide (CHX), deoxynivalenol (DON), homoharringtonine (HHT), verrucarin-A (VA); prokaryote-specific antibiotics: gentamicin (GEN), kanamycin (KAN), streptomycin (STR), tetracycline (TCN). Top: WST1 assay of mitochondrial activity and cell survival, 48 h after incubation of KO99L tPTEN −/− cells with increasing doses of the different compounds. Middle: flow cytometry analysis of cellular counts, 48 h after incubation of KO99L cells with indicated molecules. The results are representative of at least three independent experiments. Bottom: flow cytometry analysis of cell death induction by the different compounds, 48 h after incubation of KO99L cells. Data represent live cell numbers (4,6 diamidino-2-phenylindole (DAPI) negative cells). (**c**) Top: WST1 assay of mitochondrial activity and cell survival of human cell lines representative of T-ALL (Jurkat), B-ALL (NALM-6, 697) and AML (Kasumi-1, U937). Assays were performed after a 48 h incubation with increasing doses of the different E-specific antibiotics. Bottom: flow cytometry analysis of cell death induction by the different E-specific antibiotics (48 h incubation) on the leukaemic cell line panel. (**d**) Immunoblotting of murine c-myc, Mcl-1, Hsp-90 proteins and of mitochondrial Cox-2 after incubation of tPTEN −/− KO99L cells for 24 h with indicated doses of the different antibiotics. The results are representative of at least three independent experiments.

non-rotated (71,236 particles) 80S ribosome states. Both complexes contain E-site tRNA but the rotated 80S particles have eEF2, while the non-rotated particles contain no factor. 3D sorting of non-rotated states showed (after exclusion of a bad 3D class containing 1,899 particles) a class without E-site tRNA (19,026 particles) with CHX bound and the rest contained E-site tRNA (50,311 particles). To address whether the eEF2-containing complexes could have CHX bound we did an additional 3D classifications of subpopulations: we found that when eEF2 is present, the E-site tRNA is also present but there is no CHX bound; when eEF2 is absent but the E-site tRNA is present, there is also no CHX bound; there is no state with eEF2 present and tRNA absent. Thus, the CHX ligand is only present in the subpopulation that has neither E-site tRNA nor eEF2 bound. To obtain the best possible resolution on the CHX-containing 3D sub-class comprising 19,026 particles we performed 3D refinement and movie processing using uncoarsened data with a box size of 640 pixels. The post-processing procedure implemented in RELION 1.4 (ref. 19) was applied to the final maps for appropriate masking, B-factor sharpening and resolution validation to avoid over-fitting[19]; the appropriate B-factor was determined according to the procedure described[19]. The final map of the 80S/CHX complex was fine-scaled to the previous structure[9] and the resolution was estimated in Relion and IMAGIC at 0.143 FSC and half-bit criteria[21,22,23], indicating an average resolution of 3.6 Å (Supplementary Fig. 1). Local resolution estimation with ResMap[24] shows that many regions reach ~3.1 Å resolution in the 80S/CHX complex.

The cryo-EM maps were interpreted using Chimera[25], COOT[26] to derive an atomic model of the human ribosome obtained by model building and structure refinement using Phenix[27]. For this, the atomic model of the human ribosome[9] was used as starting point and refined including the CHX ligand. The atomic model was refined against the experimental cryo-EM map by iterative manual model building and restrained parameter refinement protocols (real space refinement, positional refinement, grouped B-factor refinement and simulated annealing as described in Khatter et al.[9] and Natchiar et al. (manuscript in preparation). Feature-enhanced maps[28] were used occasionally to facilitate assignment of side-chain conformations and the atomic model was then refined against the cryo-EM map as previously described[9]. The refinement process was monitored with R-factor values (25.1%/28.2% $R_{free}$; initial R 40%) to avoid over-fitting, and the entire 80S structure was refined at once by simulated annealing using parallel computing. The final atomic model comprises ~220,000 atoms (excluding hydrogens) across the 5,866 nucleotide residues and ~11,590 amino acids of the 80 proteins and the four rRNA's (28S, 5S, 5.8S and 18S; excluding certain ES rRNA which are only partially visible at the periphery of the structure probably due to conformational heterogeneity). In addition, 226 $Mg^{2+}$ ions, 32 water molecules and one CHX ligand (20 atoms) were included in the atomic model. Protein residues show well-refined geometrical parameters (allowed regions 11.8%, preferred regions 87.4% in Ramachandran plots and 0.8% of outliers). Figures were prepared using the software Chimera[25] and Pymol (DeLano, 2006).

**Cell cultures assays.** The murine KO99L cell line was established from a tPTEN −/− tumour. Human leukaemia cell lines used are: Jurkat (T-ALL: T-cell acute lymphoblastic leukaemia); NALM-6 and 697 (B-ALL: B-cell acute lymphoblastic leukaemia); Kasumi-1 and U937 (AML: Acute Myeloid Leukaemia). Cell lines were annually checked for mycoplasma infection using MycoAlert (Lonza). Cells were grown in Roswell Park Memorial Institute (RPMI) 1,640 medium (invitrogen) supplemented with 10% or 20% (KO99L) foetal calf serum and 50 units per ml penicillin, 50 mg ml−1 streptomycin and 1.0 mM sodium pyruvate (Sigma-Aldrich). Cell cultures were maintained at 37 °C under 5% $CO_2$. Peripheral blood lymphocytes were obtained from the Heparin-treated blood from healthy donors after isolation by Ficoll-Paque-Plus density gradient (Amersham Biosciences).

**Immunoblotting.** Total protein extracts solubilized in SDS sample buffer were separated on 10% SDS–polyacrylamide gel electrophoresis gels before transfering to nitrocellulose 0.45 μm membranes (GE Healthcare). Membranes were probed with antibodies obtained from Cell Signalling Technology: c-myc (#9402); Santa Cruz Biotechnology: cox-2 (sc-23983), hsp90 (sc-13119) and Rockland: mcl-1 (600-401-394), and visualized with the Pierce-ECL-substrate (ThermoScientific).

**Analysis of cell viability, proliferation and death.** In a 96-well plate, 40,000 cells per well were incubated with effectors for 48 h at 37 °C (n = 4 per experiment). For cell viability measurement, 10 μl of WST-1 (4-[3-(4iodophenyl)-2-(4-nitrophenyl)-2H-5-tetrazolio]-1,3-benzene disulfonate) reagent (Roche Diagnostics) was added to each well, incubated and the formazan production was measured at 490 nm. Cell death was assessed after staining with 4,6 diamidino-2-phenylindole followed by FACS analyses on MACSQuant or MACSQuant VYB analysers (Miltenyi Biotec). Absolute numbers of cells were also measured for each condition. All statistical data from functional assays are presented as the mean ± s.d.

**RNA and DNA isolation and quantitative real time PCR analysis.** Total RNAs from treated KO99 cells were extracted using Trizol reagent (Invitrogen) and 4 μg of RNA was reverse transcribed using GoScript reverse transcriptase (Promega). Complementary DNAs (50 ng) were amplified in triplicate on a StepOne Plus Real

Time PcR system using SYBR Green I Dye (Applied Biosystems). Primers against 36B4, Cox2 and Mcl-1 were designed using PRIMER Express Software (Applied Biosystems), and sequences are available upon request. Primers against mouse c-Myc were purchased from OriGene. Fluorescence differences were collected by the thermal cycler during each 60 °C cycle. Semi-logarithmic plots were constructed of delta fluorescence versus cycle number. The $2^{-\partial\partial Ct}$ method was used to determine the change in expression of each gene relative to 36B4 expression.

**Data availability.** Atomic coordinates and the cryo-EM map have been deposited in the Protein Data Bank and EMDB under accession codes 5LKS.pdb and EMD-4070. The data that support the findings of this study are available from the corresponding author upon request.

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

## Acknowledgements

We thank Jonathan Michalon, Remy Fritz and Romaric David for IT support, Jean-Francois Ménétret for technical support, the IGBMC cell culture facilities for HeLa cell production, and Grigory Sharov and Gabor Papai for scripting the initial work flow. This work was supported by the Centre Nationale pour la Recherche Scientifique (CNRS), Association pour la Recherche sur le Cancer (ARC), Institut National du Cancer (INCa) and the European Research Council (ERC Starting Grant No. 243296 TRANSLA-TIONMACHINERY). The electron microscope facility was supported by the Alsace Region, the Fondation pour la Recherche Médicale (FRM), the IBiSA platform programme, INSERM, CNRS and ARC, and by the French Infrastructure for Integrated Structural Biology (FRISBI) ANR-10-INSB-05-01, and Instruct as part of the European Strategy Forum on Research Infrastructures (ESFRI). J.-F.P. acknowledges the financial help of the LiSA (Lions Sport Action) association.

## Author contributions

I.H. conducted sample preparation and A.G.M. performed cryo-EM data acquisition and image processing. S.K.N. carried out structure refinement and model building and H.K. was involved in the early project. M.N. and V.I. performed functional assays. All authors analysed the data. B.P.K. and J.-F.P. supervised the project and wrote the manuscript with input from all the authors.

## Additional information

**Competing financial interests:** The authors declare no competing financial interests.

**How to cite this article**: Myasnikov, A. G. *et al.* Structure–function insights reveal the human ribosome as a cancer target for antibiotics. *Nat. Commun.* 7:12856 doi: 10.1038/ncomms12856 (2016).

