## [Peer review file · Nature Communications]

Reviewers' Comments:

Reviewer #1 (Remarks to the Author)

In this manuscript, the authors disclosed for the first time structures of human 80S ribosome in complex with E-site tRNA and the translation inhibitor cycloheximide, respectively. A comparison of the structures offers a molecular rationale for the effective competition of cycloheximide against the much larger tRNA and revealed subtle but important differences in molecular interactions with cycloheximide between human and yeast ribosome, suggesting future directions for modifying existing inhibitors to potentially improve potency. Separately, the authors tested a number of translation elongation inhibitors against a panel of leukemia cells and demonstrated anti-leukemic activity via inhibition of translation as reflected in the expression levels of c-Myc and Mcl-1.

Although the authors have previously reported the Cryo-EM structure of the human 80S ribosome and the structure of the complex between yeast ribosome and cycloheximide was reported by others, there are interesting and significant new insights revealed in this manuscript as described above. As such, the manuscript deserves consideration for publication in Nature Communications.

There are a couple of issues that the authors need to address/clarify to further improve the manuscript.

(1) In the introduction, the authors stated that "targeting the human ribosome has not yet been envisaged yet...". This is simply not true. In fact, homoharringtonine that was tested against various cancer cell lines by the author is the first approved drug targeting translation for the treatment of CML. And eukaryotic translation has been pursued as a general cancer drug target for a long time by many groups.

(2) The average resolution for the structure was reported to be 3.6 Å with some parts having higher resolution. It was unclear what is the resolution for the cycloheximide and the confidence level on the different bonding interactions described for its interaction with the ribosome.

Reviewer #2 (Remarks to the Author)

Myasnikov et al.

The manuscript entitled "Structure-function insights reveal the human ribosome as a cancer target for antibiotics" by authors Myasnikov et al. presents the results on the antibiotic CHX in two indirectly related parts of a study. One is a cryo-EM study of the human ribosome in either presence or absence of CHX along with their atomic models. The other is a study of the role of CHX in anti-proliferative dose response in leukemic cells and in interference with synthesis of c-myc and mcl-1 short-lived protein markers. This manuscript concludes that CHX specifically targets the ribosome in leukemic cells. The following comments specifically address the cryo-EM part of the study only.

The MS does not fulfill the minimum requirements for showing the evidence from which conclusions are drawn. The only two cryo-EM maps that are shown, apo, vs. CHX-bound, are uninformative as presented. Furthermore, no details of atomic modeling are presented, and the validity of structural details related to the binding of the ligand and accompanying conformational changes cannot be assessed/verified since the critical parts of the structure are not shown in the context of map density. As such, the MS is unsuited for serious consideration in this journal.

Major points in detail:

1. The cryo-EM maps are among the major results of this study, but, paradoxically, the maps are

only shown in Figure 1 in a size little larger than thumbnails. None of the panels shows the correspondence between the maps and inferred structures. The atomic models themselves lack explicit validation.

2. In Results (pages 4-5), the authors write: "The overall binding site corresponds to that observed in the yeast ribosome (8) but the ligand is slightly rotated, has the piperidine- 2,6-dione moiety flipped and exhibits an additional interaction with the 2-keto group of C4342. Specific interactions in the ligand pocket (Fig. 1c)..." . The questions are:

(a) If this is the structural base to claim that CHX specifically targets the human ribosome, Figure 1 does not show the differences in the binding of CHX with either the human ribosome or the ribosome in yeast.

(b) What is the reason and proposed mechanism for the flip?

(c) Can the slight flip result from the specific binding of CHX to the human ribosome?

3. On page 5, the authors write: "These observations illustrate the dynamic nature of ligand binding and tRNA release (which occurs even at 4°C during the ligand incubation, see methods), involving conformational changes at the ribosome level, notably helices H77/78 of the L1 stalk region which becomes disordered (Figs.1a&b)." Questions are:

(a) It is not clear what is meant here by the dynamic nature.

(b) If the authors meant the binding of CHX induces the release of tRNA, why not consider an alternative interpretation: that the binding prevents tRNA from moving into the E site?

(c) Figs. 1a&b do not show any conformational changes.

4. On page 6, the authors write: "When tested on normal peripheral blood lymphocytes (PBLs), anisomycin and verrucarin-A displayed some toxicity for normal cells (36 and 24% increase in cell death, respectively) but only at very high doses (30 and 1 µM, respectively), while CHX did not trigger cell death in normal cells compared to leukemic cell lines (Suppl. data Fig. 5)." - Are there any indications that the human ribosome is different in normal cells and in leukemic cells?

5. In Methods (page 9), the authors write: "Both complexes contain E-site tRNA but the rotated 80S particles have eEF2, while the non-rotated particles contain no factor." But the map showing eEF2 was ignored without obvious reason. It would be interesting to see if CHX is present in that map.

6. In Methods (pages 9-10), the authors write: "3D sorting of non-rotated states showed (after exclusion of bad 3D class containing 1899 particles) a class without E-site tRNA (19026 particles) with CHX bound and the rest contained E-site tRNA (50311 particles)". But they reported one resolution value only, 3.6 Angstroms ("The resolution was estimated in Relion and IMAGIC at 0.143 FSC and half-bit criteria (16, 18, 19), indicating an average resolution of 3.6 Å". It is difficult to see how these two maps, including 1899 and 50311 particles, respectively, could end up with the same resolution. An explanation is needed.

7. In Methods (page 10), the authors write: "Local resolution estimation with ResMap (20) shows that many regions reach ~3.1 Å resolution in the CHX 80S complex." - this is the map including fewer particles. It is necessary to show the local resolution map in the MS.

Minor points

1. A lot of abbreviations are used in the MS, which may not be clear to many readers. At one place, a list of the abbreviations with the full meaning for each is needed.

2. The writing at a number of places needs to be clarified. There are a number of poor, ambiguous constructions, poor word choices and missing information. For example"

(a) on page 3 "The ribosome is the molecular machinery at the heart of protein synthesis, a highly regulated function which is tightly wired to cell activation and proliferation ..."

(b) on page 7: "The tRNA complex exhibits no density for CHX...."

(c) On page 6: "The findings on the mechanism of CHX action on the human..."

(d) On page 10: "... the appropriate B-factor was determined according to ..." - what is the appropriate B-factor value?

Point-to-point response

Reviewer #1 (Remarks to the Author):

In this manuscript, the authors disclosed for the first time structures of human 80S ribosome in complex with E-site tRNA and the translation inhibitor cycloheximide, respectively. A comparison of the structures offers a molecular rationale for the effective competition of cycloheximide against the much larger tRNA and revealed subtle but important differences in molecular interactions with cycloheximide between human and yeast ribosome, suggesting future directions for modifying existing inhibitors to potentially improve potency. Separately, the authors tested a number of translation elongation inhibitors against a panel of leukemia cells and demonstrated anti-leukemic activity via inhibition of translation as reflected in the expression levels of c-Myc and Mcl-1.

Although the authors have previously reported the Cryo-EM structure of the human 80S ribosome and the structure of the complex between yeast ribosome and cycloheximide was reported by others, there are interesting and significant new insights revealed in this manuscript as described above. As such, the manuscript deserves consideration for publication in Nature Communications.

We thank the referee for the very positive feedback and the detailed insights.

There are a couple of issues that the authors need to address/clarify to further improve the manuscript.

(1) In the introduction, the authors stated that "targeting the human ribosome has not yet been envisaged yet...". This is simply not true. In fact, homoharringtonine that was tested against various cancer cell lines by the author is the first approved drug targeting translation for the treatment of CML. And eukaryotic translation has been pursued as a general cancer drug target for a long time by many groups.

We have now toned down the statement and rephrased this sentence. Homoharringtonine is indeed a very good example and illustrates the potential for cancer research of protein synthesis inhibitors. We thank the referee for these comments that help clarifying this point.

(2) The average resolution for the structure was reported to be 3.6 Å with some parts having higher resolution. It was unclear what is the resolution for the cycloheximide and the confidence level on the different bonding interactions described for its interaction with the ribosome.

As suggested, we have now estimated the local resolution and show the atomic model together with the cryo-EM. To describe this in more detail we have created an additional figure (Fig. 2) and also modified Fig. 1 to show the cryo-EM map of the L1 region.

Reviewer #2 (Remarks to the Author):

The manuscript entitled "Structure-function insights reveal the human ribosome as a cancer target for antibiotics" by authors Myasnikov et al. presents the results on the antibiotic CHX in two indirectly related parts of a study. One is a cryo-EM study of the human ribosome in either presence or absence of CHX along with their atomic models. The other is a study of the role of CHX in anti-proliferative

dose response in leukemic cells and in interference with synthesis of c-myc and mcl-1 short-lived protein markers. This manuscript concludes that CHX specifically targets the ribosome in leukemic cells. The following comments specifically address the cryo-EM part of the study only.

The MS does not fulfill the minimum requirements for showing the evidence from which conclusions are drawn. The only two cryo-EM maps that are shown, apo, vs. CHX-bound, are uninformative as presented. Furthermore, no details of atomic modeling are presented, and the validity of structural details related to the binding of the ligand and accompanying conformational changes cannot be assessed/verified since the critical parts of the structure are not shown in the context of map density. As such, the MS is unsuited for serious consideration in this journal.

We thank the referee for the constructive comments and for suggesting to show more details on the cryo-EM map and the derived atomic model. This is now addressed in detail by modifying Fig. 1 and by creating a new figure (Fig. 2) as described below.

Major points in detail:

1. The cryo-EM maps are among the major results of this study, but, paradoxically, the maps are only shown in Figure 1 in a size little larger than thumbnails. None of the panels shows the correspondence between the maps and inferred structures. The atomic models themselves lack explicit validation.

We have now added several images to show the cryo-EM map and the derived atomic model. To describe this in more detail we have created an additional figure (Fig. 2) which shows the atomic model together with the cryo-EM of the ligand-binding pocket region and a local resolution estimation. Moreover, we modified Fig. 1 to also show the cryo-EM map of the L1 region and describe the accompanying conformational changes. In addition, we provide an additional figure in the Suppl. Data (Suppl. Data Fig. 2) to show the overall quality of the cryo-EM map and the atomic model. Finally, the geometric parameters of the derived atomic model show a very good model quality as described in the methods section, further refined while the manuscript was under review and also improved over our previously published human 80S ribosome structure (Khatter et al., 2015).

2. In Results (pages 4-5), the authors write: "The overall binding site corresponds to that observed in the yeast ribosome (8) but the ligand is slightly rotated, has the piperidine- 2,6-dione moiety flipped and exhibits an additional interaction with the 2-keto group of C4342. Specific interactions in the ligand pocket (Fig. 1c)...". The questions are:
(a) If this is the structural base to claim that CHX specifically targets the human ribosome, Figure 1 does not show the differences in the binding of CHX with either the human ribosome or the ribosome in yeast.

We now show a comparison of the human and yeast ribosome structures with CHX (new Fig. 2d)

(b) What is the reason and proposed mechanism for the flip?

The reason for the flip could be the presence of a Mg²⁺ ion that is absent in the yeast ribosome CHX complex. We have added this in the text now.

(c) Can the slight flip result from the specific binding of CHX to the human ribosome?

This is difficult to answer. A possibility is the specific presence of the Mg^{2+} ion that may have an influence on the positioning of the piperidine-2,6-dione moiety as suggested above.

3. On page 5, the authors write: "These observations illustrate the dynamic nature of ligand binding and tRNA release (which occurs even at 4{degree sign}C during the ligand incubation, see methods), involving conformational changes at the ribosome level, notably helices H77/78 of the L1 stalk region which becomes disordered (Figs.1a&b)." Questions are:

(a) It is not clear what is meant here by the dynamic nature.

(b) If the authors meant the binding of CHX induces the release of tRNA, why not consider an alternative interpretation: that the binding prevents tRNA from moving into the E site?

This is indeed what was meant: CHX induces the release of the tRNA, and inversely once CHX is bound it prevents the tRNA from rebinding; moreover, during elongation it would block tRNA translocation from the P- to the E-site. We thank the referee for this clarification. We have modified the text accordingly.

(c) Figs. 1a&b do not show any conformational changes.

As suggested, we have now modified Fig. 1 to show the conformational changes of the L1 region in more detail (cryo-EM map and atomic model).

4. On page 6, the authors write: "When tested on normal peripheral blood lymphocytes (PBLs), anisomycin and verrucarin-A displayed some toxicity for normal cells (36 and 24% increase in cell death, respectively) but only at very high doses (30 and 1 μ M, respectively), while CHX did not trigger cell death in normal cells compared to leukemic cell lines (Suppl. data Fig. 5)." - Are there any indications that the human ribosome is different in normal cells and in leukemic cells?

We think that in these experiments the primary effect comes from the different sensitivity of cancer cells due to the stronger dependency on protein synthesis as compared to normal cells; we have reformulated the text to clarify this point. Many gene expression data show variations in mRNAs for ribosomal proteins between normal and cancer cells, opening the possibility that "cancer" ribosomes could differ in their structure and therefore in their function compared to "normal" ones. Identifying differences between ribosomes in normal cells and in leukemic cells is indeed an interesting aspect to analyse in future studies.

5. In Methods (page 9), the authors write: "Both complexes contain E-site tRNA but the rotated 80S particles have eEF2, while the non-rotated particles contain no factor." But the map showing eEF2 was ignored without obvious reason. It would be interesting to see if CHX is present in that map.

To address this point we did an additional analysis of the cryo-EM data through 3D classifications of subpopulations. We found that when eEF2 is present, the E-site tRNA is also present but there is no CHX bound. When eEF2 is absent but the E-site tRNA is present, there is also no CHX bound. There is no state with eEF2 present and tRNA absent. Thus, the CHX ligand is only present in the subpopulation that has neither tRNA nor eEF2 bound. We have clarified this in the methods section now.

6. In Methods (pages 9-10), the authors write: "3D sorting of non-rotated states showed (after exclusion of bad 3D class containing 1899 particles) a class without E-site tRNA (19026 particles) with

CHX bound and the rest contained E-site tRNA (50311 particles)". But they reported one resolution value only, 3.6 Angstroms ("The resolution was estimated in Relion and IMAGIC at 0.143 FSC and half-bit criteria (16, 18, 19), indicating an average resolution of 3.6 Å". It is difficult to see how these two maps, including 1899 and 50311 particles, respectively, could end up with the same resolution. An explanation is needed.

The CHX complex comprises 19026 particles, not 1899 (the latter is the number of discarded particles within one of the subpopulations). The structure of the tRNA/80S complex is the same as published previously, hence we did not describe the processing to the same extent as for the CHX complex. Resolution values refer only to the CHX complex. The tRNA-containing 80S complex could be refined more, but we focused our attention on the 80S/CHX complex to obtain the best possible resolution of the ligand complex. We have clarified this in the text now.

7. In Methods (page 10), the authors write: "Local resolution estimation with ResMap (20) shows that many regions reach ~3.1 Å resolution in the CHX 80S complex." - this is the map including fewer particles. It is necessary to show the local resolution map in the MS. As suggested, we now show the local resolution map (new Fig. 2b).

Minor points

1. A lot of abbreviations are used in the MS, which may not be clear to many readers. At one place, a list of the abbreviations with the full meaning for each is needed.

Abbreviations are introduced when they appear in the text. We have now added also a list of abbreviations.

2. The writing at a number of places needs to be clarified. There are a number of poor, ambiguous constructions, poor word choices and missing information. For example"

(a) on page 3 "The ribosome is the molecular machinery at the heart of protein synthesis, a highly regulated function which is tightly wired to cell activation and proliferation ..."

(b) on page 7: "The tRNA complex exhibits no density for CHX..."

(c) On page 6: "The findings on the mechanism of CHX action on the human..."

(d) On page 10: "... the appropriate B-factor was determined according to ..." - what is the appropriate B-factor value?

These sections have been rephrased accordingly.

Reviewers' Comments:

Reviewer #1 (Remarks to the Author)

All my concerns have been addressed by the authors. I recommend acceptance of the manuscript.

Reviewer #3 (Remarks to the Author)

I was asked to provide brief comments about the revised paper from the technical view point. The Klaholz group was the first to report the atomic structure of a human ribosome (Nature 2015). In the current paper, his group extends from their Nature study and reports the structures of human 80S ribosome in complex with E-site tRNA and the translation inhibitor cycloheximide. To my knowledge, this is the first report of the human ribosome with a translation inhibitor. As with the previously published Nature structure, I consider the structure determination done properly and the structural features reported here are consistent with the stated resolution of about 3.6 Å. The structures are reported in a manner supported by the data.

The revised paper and the authors' responses both adequately addressed the issues raised by the previous reviewers.

The publication of this paper should attract wide-spread interest from both the structural biologists and pharmaceutical investigators.